# Cryptic functional diversity within a grass mycobiome

**Cedric Ndinga-Muniania**[1,2]*, **Nicholas Wornson**[3,4], **Michael R. Fulcher**[5], **Elizabeth T. Borer**[2], **Eric W. Seabloom**[2], **Linda Kinkel**[2,4], **Georgiana May**[2]

**1** Plant and Microbial Biology Graduate Program, University of Minnesota, St. Paul, Minnesota, United States of America, **2** Department of Ecology, Evolution and Behavior, University of Minnesota, St. Paul, Minnesota, United States of America, **3** School of Statistics, University of Minnesota, Minneapolis, Minnesota, United States of America, **4** Department of Plant Pathology, University of Minnesota, Saint Paul, Minnesota, United States of America, **5** Foreign Disease–Weed Science Research Unit, United States Department of Agriculture, Frederick, Maryland, United States of America

* nding001@umn.edu

**Data Availability Statement:** All relevant data are within the paper and its Supporting Information files. All ITS sequences are available from the Genbank database.

## Abstract

Eukaryotic hosts harbor tremendously diverse microbiomes that affect host fitness and response to environmental challenges. Fungal endophytes are prominent members of plant microbiomes, but we lack information on the diversity in functional traits affecting their interactions with their host and environment. We used two culturing approaches to isolate fungal endophytes associated with the widespread, dominant prairie grass *Andropogon gerardii* and characterized their taxonomic diversity using rDNA barcode sequencing. A randomly chosen subset of fungi representing the diversity of each leaf was then evaluated for their use of different carbon compound resources and growth on those resources. Applying community phylogenetic analyses, we discovered that these fungal endophyte communities are comprised of phylogenetically distinct assemblages of slow- and fast-growing fungi that differ in their use and growth on differing carbon substrates. Our results demonstrate previously undescribed and cryptic functional diversity in carbon resource use and growth in fungal endophyte communities of *A. gerardii*.

## Introduction

The growing understanding that microbiomes play important and diverse roles in host responses to changing environments has stimulated research in the taxonomic and functional diversity of microbiome communities [1–3]. Found in association with virtually all eukaryotic hosts, microbial symbionts maintain a wide variety of interactions with their host, ranging from beneficial to antagonistic, and affect host responses to the environment [4–8]. While next generation sequencing technologies have provided key insights into the taxonomic and functional genomic diversity of microbiomes [9–18], the processes generating and maintaining functional diversity in communities of many symbiotic organisms need further investigation. We investigated patterns of carbon resource use and growth of fungal endophytes associated with the prairie grass species, *Andropogon gerardii*, to better understand the evolutionary origins of taxonomic and functional diversity in this key group of plant symbiotic organisms.

**Funding:** This study was supported by a National Science Foundation (NSF) Macrosystems Biology grant (NSF-DEB 00037623) to co-PIs EB, ES, LK, GM. Support was also provided from the NSF Long Term Ecological Research (NSF-DEB-1234162 and NSF-DEB-1831944 to Cedar Creek LTER) and Research Coordination Network (NSF-DEB-1042132) programs. Support to CNM was provided by the NSF-DEB, a Dissertation Fellowship from the Graduate School at University of Minnesota, and from the Graduate Program in Plant and Microbial Biology. The funders had no role in study design, data collection and analysis, decision to publish, or preparation of the manuscript.

**Competing interests:** The authors have declared that no competing interests exist.

Patterns of resource use and growth largely define the ecological niche of microbiome species and inform processes structuring their communities [19–22]. For example, in saprobic communities, the most abundant fungal species on recalcitrant substrates such as lignin and hemicellulose are those able to break down these substrates as a main source of carbon [23–26]. Like saprobic communities, differing patterns of resource use by symbiotic microbes may also define different niches within the host [27] and mediate biotic interactions among co-occurring symbionts [28–33]. For example, in plant symbiotic communities such as ectomycorrhizal fungi, the differing use of carbon and nitrogen resources by co-occurring taxa delimits differing niches, reducing competition and promoting coexistence of diverse species within these communities [34–37]. In contrast, overlap in resource use can intensify competition among co-occurring microbial symbionts, potentially contributing to lower diversity [38]. Understanding the functional diversity of resource use in microbial symbiont species will improve our understanding of how niche differences and biotic interactions affect the assembly of symbiotic microbiomes.

Community phylogenetic approaches complement ecological studies as they inform the extent to which observed patterns of taxonomic and functional diversity may result from a shared evolutionary history [39, 40]. For microbial symbionts such as the fungal endophytes we study here, the endophytic trophic mode has evolved multiple times across the phylum Ascomycota [41, 42] contributing to the high level of taxonomic diversity observed in these communities [43, 44]. Although community phylogenetic studies often focus at or above the species level, adaptation to local environment and to biotic interactions may give rise to trait variation within and among closely related species and contribute to diversity [45–48]. In contrast, the constraints imposed by stressful environments may lead to less diverse communities [49, 50], especially when variation in the traits associated with tolerance of these conditions has evolved in only a few lineages and is phylogenetically constrained [51, 52]. For example, the low diversity in fungal communities associated with lignin-rich substrates [26, 53] may be explained in part by the observation that the ability to efficiently break down lignin has mainly evolved within the fungal class Agaricomycetes [54, 55]. Thus, understanding the phylogenetic scale at which resource use and growth traits are conserved will provide insight into evolutionary processes generating taxonomic and functional diversity in host-associated microbiomes.

We evaluated the taxonomic diversity and resource use and growth traits of fungal endophyte communities associated with the native grass *A. gerardii* (big bluestem), which is widespread throughout the United States Great Plains region. Fungal endophytes are dominant members of foliar microbiomes and constitute hyper-diverse communities living inside of plant tissues without causing apparent disease symptoms [43, 56–58]. To capture the taxonomic and functional diversity of *A. gerardii* endophytes, we sampled leaves of plants growing under field conditions of differing nutrient conditions and isolated fungi using two methods, leaf sectioning and leaf maceration [59, 60]. We determined differences in use of carbon resources and growth on those resources, and used community phylogenetic approaches to evaluate the extent to which similarity of functional traits among isolates might be due to shared evolutionary history [61–63]. Our results demonstrate that fungal communities within *A. gerardii* leaves harbor two previously undescribed, functionally and phylogenetically distinct assemblages of slow- and fast-growing fungi and suggest that both stochastic and deterministic processes underpin the diversity of fungal symbionts observed within a host.

## Materials and methods

### Study site

Leaf samples were collected from *A. gerardii* plants growing in experimental plots established as part of the Nutrient Network distributed experiment (NutNet; [64]) at the

University of Minnesota Cedar Creek Ecosystem Science Reserve (CCSER; Latitude 45.4 N, Longitude 93.2 W) in a successional agricultural field with sandy, low fertility soils. The NutNet experimental site includes 5 x 5 m plots of nutrient addition treatments (NPK+; 10 g N m$^{-2}$yr$^{-1}$ as urea, 10 g P m$^{-2}$yr$^{-1}$ as Ca(H$_2$PO$_4$)$_2$, 10 g K m$^{-2}$yr$^{-1}$ as K$_2$SO$_4$, and 100 gm$^{-2}$yr$^{-1}$ of a micronutrient mix) and Control plots without nutrient treatment [65, 66], in a randomized block design. The experimental plots were established in 2007 and treatments have been subsequently applied annually. The impacts of the NPK+ treatment on foliar endophyte communities have been reported in other studies which show shifts in community composition due to fertilization treatments within sites [65, 67–70]. Here, we evaluate the functional and taxonomic diversity of fungal endophytes occurring across the NPK+ and Control plots.

## Tissue sampling and endophytic fungal isolation

Sampling was conducted in August 2015 at peak biomass for *A. gerardii*. One fully emerged *A. gerardii* leaf without disease symptoms was collected from each of three plants in each of two replicate Control and two NPK+ plots for total of 12 leaves (3 plants X 2 replicate plots X 2 treatments). Leaf samples were individually bagged, placed on ice, returned to the lab, and processed within 48 hours following the protocol of Arnold et al. [43]. Briefly, leaves were rinsed in water and surface sterilized by sequential rinses in 75% ethanol (1 min), 50% commercial bleach (1 min), 75% ethanol (1 min) and sterile dH$_2$O (1 min). Surface sterilized leaves were each divided into three approximately equal length segments, and one segment was randomly assigned to each of two different isolation methods. For the *leaf sectioning* isolation method, surface sterilized leaf segments were cut into 5 mm$^2$ pieces and incubated on 2% potato dextrose agar (PDA; Difco, Sparks, MD, USA) in 1.5 mL Eppendorf tubes. For the *leaf maceration* method, surface sterilized leaves were cut into small sections as above and macerated in 3 mL PBS buffer (0.01 M phosphate, 0.137 M NaCl, 2.7 mM KCl, pH 7.3) for 1 min using the FSH-125 homogenizer (Thermo Fisher Scientific, Waltham, MA, USA). The homogenized mixture was serially diluted and plated at three dilutions ($10^{-2}$, $10^{-4}$, $10^{-6}$) onto malt extract agar and water agar, and incubated at 22 ˚C under ambient light/dark conditions. Single fungal colonies were randomly collected as they appeared and transferred to small 2% PDA slants. Isolates from both methods were monitored for growth, and after 6 months, covered with sterile dH2O for storage at room temperature. Both approaches generate single fungal isolates, which were further verified by sequencing.

## DNA extraction and ITS amplification

Total genomic DNA was extracted using the REDExtract-N-Amp$^{TM}$ Tissue PCR Kit (Sigma-Aldrich, St. Louis, MO, USA) from all fungal cultures obtained by both isolation methods. To identify individual isolates, the ~ 650 bp region of the rDNA sequence spanning the Internal Transcribed Spacer Region (ITS) and including ~ 160 bp of the Large Subunit Region (LSU) was PCR amplified as a single amplicon product using the primer pairs ITS1F [71] and LR3 [72]. The PCR products were evaluated via gel electrophoresis in 1.2% agarose gel, primers removed using ExoSap-IT Product Cleanup Reagent (Thermo Fisher Scientific, Waltham, MA, USA) and single-strand Sanger sequencing was performed using the ITS1F primer (Genewiz; South Plainfield, NJ, USA). Sequence chromatograms were inspected manually in Geneious (Mac Version 5.5.6), and sequences shorter than 200 bp, of low quality (< 70% readable peaks), or those with multiple overlapping peaks indicative of multiple fungal taxa, were removed from subsequent analyses as were the isolates.

## Sequence analysis and OTU assignment

Following [73], fungal ITS-partial LSU sequences were clustered into Operational Taxonomic Units (OTU) at 97% similarity against the UNITE fungal database [74] using USEARCH [75] in QIIME v1.8.0 [76]. We used the Evolutionary Placement Approach (EPA) within the Tree-Based Alignment Selector (T-BAS) toolkit 2.1 [77, 78] to place the ITS sequences onto a six-locus reference fungal phylogeny [77–80] and assign taxonomy at the lowest taxonomic rank of the best match. All culture sequences were deposited in GenBank under accession numbers OQ979626—OQ980092.

## Resource use and growth

We first used stratified random sampling to select a subset of fungal cultures for evaluation of resource use and growth on Biolog SF-P2 plates (https://www.biolog.com/) with 95 different carbon resources (S1 Table). Taxon identity and isolation method were designated as strata and isolates chosen randomly within those strata until 10 isolates per leaf and isolation method were obtained, giving a total of 240 isolates (10 isolates × 2 isolation methods × 12 leaves).

The 240 selected isolates were grown on Malt Extract or V8 agar (Difco, Sparks, MD, USA) for at least two weeks to induce sporulation. Following [32], spores were collected using sterile cotton swabs, suspended in sterile $dH_2O$, and spore adjusted to an optical density of 0.22 at 590 nm ($OD_{590}$). These were diluted 10-fold with 0.2% carrageenan, 100 uL inoculated into each Biolog well, incubated at 28°C for 3 days, and the $OD_{590}$ recorded (BioTek Microplate reader; Winooski, VT, USA). For each isolate and carbon resource, we calculated the *standardized growth* as the difference in $OD_{590}$ compared to that of the water control well (standardized growth = carbon resource $OD_{590}$ –water control $OD_{590}$) at 3 days. Standardized growth values < 0.005 (the photometric accuracy of the microplate reader) were considered as no growth. Thus, for each isolate and Biolog carbon resource, we evaluated resource use as a binary character (1, growth; 0, no growth) and calculated the standardized growth at $OD_{590}$.

## Statistical analyses

All statistical analyses were conducted using R 3.5.1 [81]. Differences in mean number of fungal OTUs per leaf, niche width, and standardized growth were evaluated using student's t-tests (R stats package [81]). We used a non-parametric Fligner-Killeen median test (*fligner.test*, R stats package [81]) to evaluate differences in variances of niche widths [82].

## Phylogenetic patterns of growth traits

To investigate the phylogenetic distribution of growth traits (standardized growth on each Biolog carbon resource, a quantitative trait) we used the *phylosignal* function in the picante R package [83]. Because *phylosignal* requires a fully resolved phylogeny, polytomies on the phylogeny were resolved by randomly transforming multichotomies into a series of dichotomies [84] (*multi2di* function of the ape R package [85]). Phylosignal evaluates Blomberg's K statistic [61] with two models, Brownian Motion (BM) and a Random Tips (RT). The BM model generates a null distribution of trait values in descendants relative to the ancestral state by randomly assigning small changes of trait values at each node in the phylogeny [61, 86]. The Blomberg's K statistic is the standardized ratio of the observed mean square error of the phylogenetically corrected mean of trait states, divided by the expected mean squared error of those traits [61]. For each trait, a value K = 1 indicates that the observed and expected trait distributions under BM are similar, K <1 indicates that trait evolution is more labile than expected under the BM model, and a K >1 indicates that traits are more conserved than expected under

the BM model. The RT model evaluates similarity in traits values among closely related taxa by randomly shuffling taxon labels across the tips of the phylogeny (999 permutations) to generate a null distribution of K values. A calculated K value larger than 95% of values in the null distribution (one-tailed test, *P = 0.05*) demonstrates greater similarity in a trait among closely related taxa than expected by chance. To determine the phylogenetic level at which growth traits were conserved, the BM and RT approaches were applied to a phylogeny that included all isolates for which growth traits were evaluated, and separately, for subtrees of taxa representing each of the most common fungal classes and orders. Subtrees were constructed at each phylogenetic level (class or order) using the *keep.tip ()* function of the ape R package [85].

## Results

### Fungal isolation, OTU assignment and richness

A total of 531 fungal cultures were isolated by the two methods. In addition to visual assessment of colony morphology, we verified single taxon cultures as those having high quality single DNA sequences (lack of multiple nucleotide peaks) and after discarding low quality sequences, obtained a total of 476 DNA sequences. The mean sequence length of these 476 sequences was 595 bp and covered the ITS region and a small portion of the LSU region (mean 156 bp). The 476 sequences clustered into 82 OTUs at 97% similarity, and of those, 45 OTUs (9.5% of total sequences) were represented by only one culture (singletons). A high fraction of OTUs represented only once in the sample suggests communities comprised of few abundant and many rare taxa, as previously observed for *A. gerardii* [68]. Most of the isolates obtained belonged to the Phylum Ascomycota (73 of 82 OTUs; 98% of sequences). Of the remaining OTUs, 7 were assigned to Basidiomycota (1.6% of sequences) and 2 (0.4% of sequences) could not be assigned to a taxonomic level below Kingdom Fungi. Leaf sectioning recovered a somewhat greater richness of OTUs per leaf (8.6 ± 1.8) than did leaf maceration (6.6 ± 2.3) (*t-test*, *P* = 0.028) and the taxa recovered by the two isolation methods largely belong to different phylogenetic lineages (S1 Fig).

### Resource use and growth

To evaluate the functional diversity of isolates, we first randomly sampled 10 isolates from each leaf and by the two isolation methods (see Materials and Methods; 10 isolates/leaf x 12 leaves x 2 isolation methods; 240 isolates total). The 240 selected isolates represent 61 of all 82 OTUs (74%) and included all the most common OTUs. Results show that Biolog resource use (standardized $OD_{590} > 0.005$) varied greatly among isolates, and to minimize effects of zero-weighted growth values, we calculated mean standardized growth for each isolate based on the 62 Biolog carbon resources used by at least 50% of all isolates (S2 Fig, S1 Table). Mean standardized growth of isolates recovered using leaf sectioning was significantly lower ($OD_{590}$ = 0.04 ± 0.03) than the mean standardized growth of isolates recovered using leaf maceration ($OD_{590}$ = 0.29 ± 0.14) (*t-test* p< 0.001; Fig 1).

Using a finite mixture model [87], we characterized differences in the distributions of growth values of isolates and determined a cut-off value at 0.12 $OD_{590}$ to distinguish the slow- and fast-growing fungi (Slow: 95% CI = *0.03–0.05*, Fast: 95% CI = *0.27–0.31;* Fig 1). To evaluate the sensitivity of the designation of slow- and fast-growing types to the choice of 62 resources, we calculated mean standardized growth for all 95 Biolog carbon resources, and for only those resources used by each isolate (growth efficiency; see [32]). Results show that bimodal distributions of mean standardized growth are obtained using all three measures (S3 Fig). Consequently, we distinguish slow- or fast-growing types based on the mean standardized growth on 62 Biolog carbon resources and a 0.12 $OD_{590}$ cut-off value. Below, we refer to

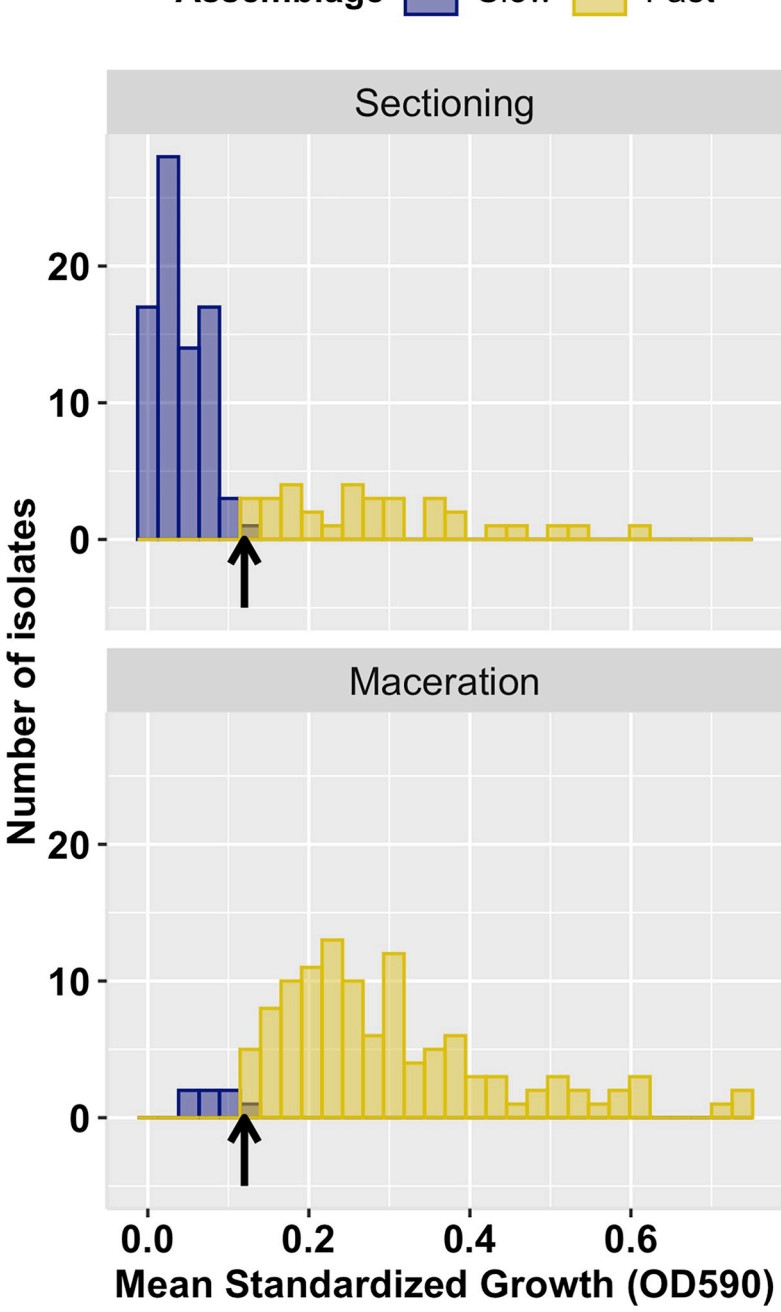

**Fig 1. Distribution of mean standardized growth values among fungal isolates obtained by leaf sectioning and maceration.** Mean standardized growth = mean (measured $OD_{590}$ –water control $OD_{590}$) on 62 Biolog carbon resources after 3 days (see Materials and Methods). Slower-growing fungi (blue bars) were primarily obtained by leaf sectioning (top panel) and faster-growing fungi (gold bars) were primarily obtained by leaf maceration. We fit a finite mixture model to derive the cut-off value of 0.12 $OD_{590}$(black arrow) to distinguish the distributions of slow- ($< 0.12$ $OD_{590}$) and fast- growing ($> 0.12$ $OD_{590}$) fungi.

slow- and fast-growing *assemblages* to distinguish these sub-communities that co-occur within each leaf.

## Niche width

Results show that slow-growing isolates exhibit a significantly lower mean but higher variance in niche width (46 ± 25) than do fast-growing fungi (82 ± 12; Fligner-Killeen median test, p< 0.001; Fig 2). Together, the results for resource use and growth demonstrate that fungal endophyte communities in *A. gerardii* leaves are comprised of slow- and fast-growing assemblages that were differentially recovered by our two isolation methods and that differ in their patterns of resource use and growth.

## Taxonomic composition and phylogenetic structure

We next evaluated differences in taxonomic composition and phylogenetic structure of fungi comprising the slow- and fast-growing assemblages. Results show that the slow- and fast-growing assemblages share only 5 of 61 OTUs and that these represent only a few assigned species (*Xylaria hypoxylon*, *Lachnum virgineum*, *Pyrenophora phaeocomes*) (Fig 3, S2 Table).

The phylogenetic structure of the two assemblages differs as well. We compared the numbers of isolates belonging to the slow- and fast-growing assemblages and that were assigned at each level of phylogenetic classification (Fig 3) using a pairwise Fisher exact test ([88], S3 Table). At the class level, slow-growing fungi occur more frequently than do fast-growing fungi in the class Sordariomycetes, whereas fast-growing fungi occur more frequently than do slow-growing fungi in the class Eurotiomycetes. Both the slow- and fast-growing fungi occur

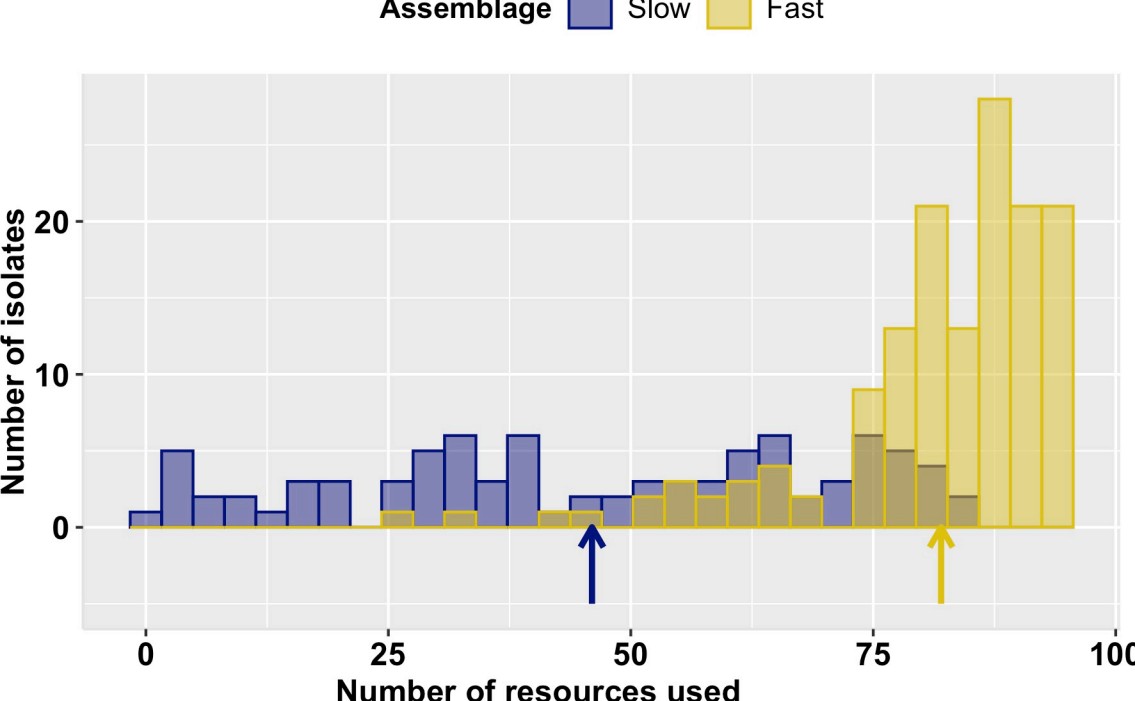

**Fig 2. Distribution of niche widths in slow- and fast-growing assemblages.** Slow-growing fungi (blue bars) had lower mean but higher variance in niche width (mean = 46, sd = 25) than did fast-growing fungi (gold bars; mean = 82, sd = 12). Niche width is the number of resources used by an isolate (standardized growth > 0.005 $OD_{590}$ after 3 days). Arrows show mean niche width of slow- (blue) or fast- (gold) growing assemblages.

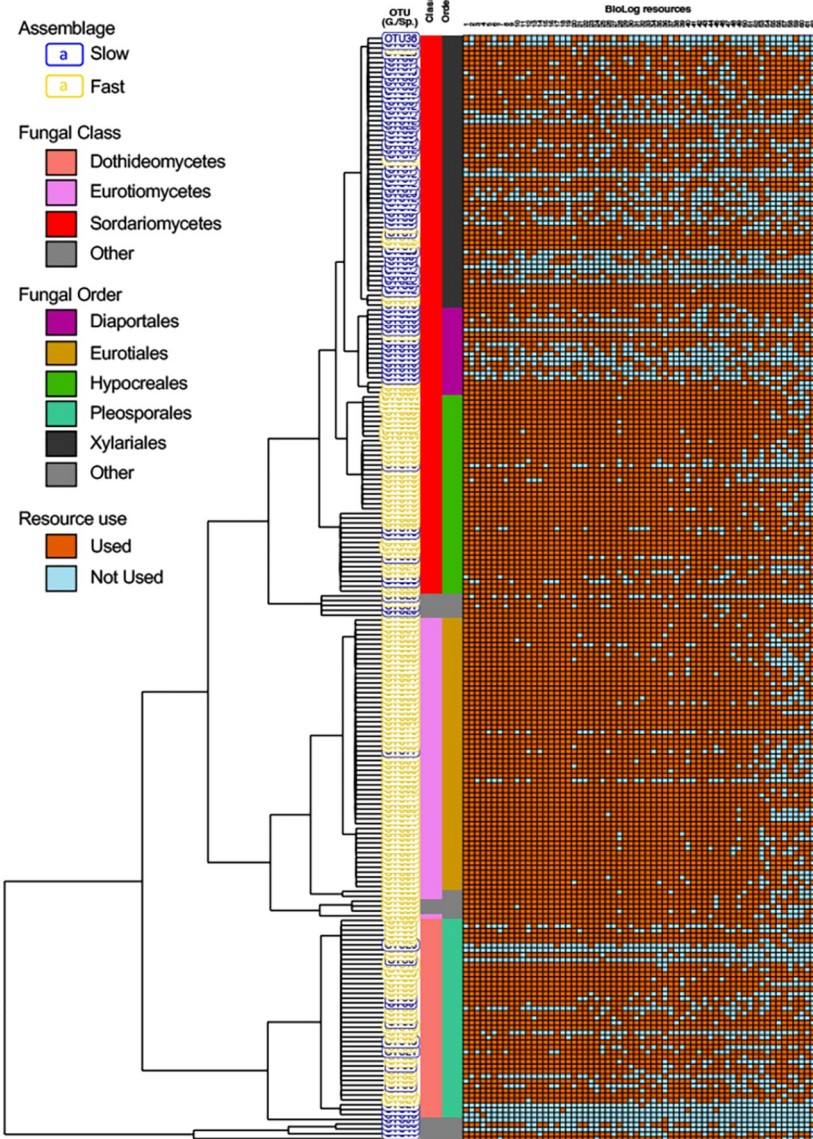

**Fig 3. Phylogenetic structure of resource use.** The OTU labels are color-coded (slow: blue; fast: gold) for each isolate evaluated on Biolog C resource plates. Columns represent fungal class and order ("Other" represents classes or orders with a relative abundance ≤2%). Resource use (standardized growth > 0.005 OD$_{590}$) is shown in the grid to the right (used: dark orange, not used: blue). Within the class Sordariomycetes, taxa in the orders Diaporthales and Xylariales are most often slow-growing whereas taxa in order Hypocreales are often fast-growing. Taxa in the Eurotiales (class Eurotiomycetes) are often fast-growing. Both assemblages have members in the class Dothidiomycetes (see S3 Table).

at similar frequency in the class Dothidiomycetes. At the order level within the Sordariomycetes, slow-growing fungi occur more frequently than do fast-growing fungi in the orders Diaporthales and Xylariales while the fast-growing fungi occur more frequently than do slow-growing fungi in the order Hypocreales. The observation that fungi in the order Xylariales are often darkly pigmented and are also often slow-growing suggests that pigmentation does not upwardly bias the standardized OD$_{590}$ growth measure. Within the class Eurotiomycetes, most isolates belong to the order Eurotiales and within the class Dothideomycetes, most isolates belong to the order Pleosporales, and results are similar at the class and order level.

Further inspection of the phylogenetic results shows that fungal orders include large "combs" of closely related OTUs assigned to only two or three named species (Fig 3; S2 Table). Within Sordariomycetes, the order Xylariales included eleven OTUs assigned to one species (*Xylaria hypoxylon*), the Diaporthales included three OTUs assigned to two species (*Gnomonia gnomon* and *Diaporthe eres*), and the Hypocreales included thirteen OTUs assigned to two species (*Fusarium graminearum* and one unidentified species). Similarly, within the Eurotiomycetes, the order Eurotiales included ten OTUs assigned to two species (*Aspergillus fumigatus* and *Aspergillus nidulans*) and within Dothideomycetes, the order Pleosporales included eleven OTUs assigned to two species (*Pyrenophora phaecomes* and *Pleospora herbarum*; S2 Table). Together, these results demonstrate that while taxa constituting slow- and fast-growing assemblages are largely phylogenetically distinct, representatives of the two assemblages are found at each phylogenetic level.

## Evolution of variation in growth traits

We used community phylogenetic approaches to evaluate the extent to which growth traits were conserved or labile over the phylogenetic history of the fungi, using Blomberg's K [61]. First considering results under the BM model and over the entire phylogeny (Fig 3), standardized growth on only 2 of the 95 resources (2%; Thymidine-5'-Monophosphate and Dextrin) showed slightly greater similarity among related fungi (K = 1.01 and K = 1.05, respectively) than expected under the BM model (Table 1, S4 Table). This result shows that growth on almost all Biolog carbon resources is labile over the evolutionary history of the Phylum Ascomycota. Not surprisingly, at the fungal class level, taxa within classes show greater similarity of growth traits than across the entire phylogeny. Classes vary in conservation of traits with taxa within the Sordariomycetes demonstrating a greater number of resources for which growth traits are conserved (64) compared to taxa within Eurotiomycetes (2) or Dothideomycetes (24) (Table 1). However, the strength of this inference is limited because the classes Eurotiomycetes

**Table 1. Phylogenetic patterns in fungal growth traits.**

| | Brownian Motion | | Random Tip | | Number of isolates | |
|---|---|---|---|---|---|---|
| | $K_{obs}$† | Number of resources for which K > 1‡ | Number of resources for which P < 0.05‖ | $P$§ | Slow | Fast |
| **Entire phylogeny** | 0.31–1.05 | 2 | 74 | 0.00–0.83 | 93 | 147 |
| **Dothideomycetes** | 0.81–1.20 | 24 | 4 | 0.00–0.91 | 12 | 29 |
| **Eurotiomycetes** | 0.70–1.14 | 2 | 13 | 0.00–0.74 | 1 | 57 |
| **Sordariomycetes** | 0.52–2.28 | 64 | 75 | 0.00–0.93 | 69 | 46 |
| **Hypocreales** | 0.68–1.70 | 35 | 58 | 0.00–1.00 | 6 | 35 |
| **Diaporthales** | 0.82–2.70 | 66 | 52 | 0.00–0.98 | 14 | 4 |
| **Xylariales** | 0.71–3.01 | 44 | 26 | 0.00–0.96 | 48 | 5 |

Summary of results for conservation of growth traits under Brownian Motion (BM) and Random Tip (RT) models. Growth on each of 95 Biolog carbon resources was assessed, and the conservation of these growth traits assessed using Blomberg's K statistic [61]. Under the BM model, K > 1 indicates the growth trait is more conserved at nodes across the phylogeny than under null expectations. Under the RT model, P < 0.05 indicates that traits are significantly more similar among closely related taxa than expected under null expectations. Results for analyses of growth traits on individual Biolog carbon resources are reported in S4 Table.

†: Range of observed K statistic values for all 95 Biolog carbon resources at each phylogenetic level: the entire phylogeny, the most common classes, and the most common orders.

‡: Number of growth traits for which K > 1 out of 95 total (BM model)

‖: Number of growth traits for which K is significantly greater (P < 0.05) than expected under the null distribution of K values (RT model).

§: Range of P values obtained for all 95 Biolog resources (RT model).

Also reported are the numbers of isolates from the slow- and fast-growing assemblages included at each phylogenetic level.

and Dothideomycetes each include only one group of closely related taxa (large comb, Fig 3). At the level of orders, we examined subtrees only for the three orders within the class Sordariomycetes because the Eurotiomycetes and Dothidiomycetes were each represented by taxa in one order. Standardized growth on 35 resources in Hypocreales, on 66 resources in Diaporthales and on 44 resources in Xylariales demonstrated greater similarity among taxa within order than expected by under the BM model (Table 1, S4 Table). Our ability to compare the extent of conservation in growth traits among orders is limited because of the differing numbers of clades within each order. Nonetheless, along with the observation that the Diaporthales and Xylariales are dominated by slow-growing taxa and the Hypocreales are dominated by fast-growing taxa, the BM model results show that growth traits tend to be most similar among taxa within orders and classes.

We investigated variation in growth traits among more closely related taxa using results under the RT model. Over all taxa represented in the phylogeny, we found that growth traits were more similar than expected by chance on 74 out of the 95 resources (p< 0.05; Table 1, S4 Table). The greater level of similarity in growth traits under the RT model compared to the BM model for the entire phylogeny shows that while growth on these resources is labile over long periods of time, more closely related taxa have similar resource use and growth patterns. At the class level, growth traits were more similar than expected under the RT model among taxa in the Sordariomycetes (75), compared taxa in the Eurotiomycetes (13) or the Dothideomycetes (4). The apparent lower level of conservation in the latter two classes may be in part due to their phylogenetic structure with few combs of closely related taxa and because the Dothideomycetes harbor both slow- and fast-growing fungi. At the order level within the Sordariomycetes, growth traits were more similar among closely related taxa than expected by chance on 58 resources in the Hypocreales, 52 resources in the Diaporthales and 26 resources in the Xylariales (Table 1, S4 Table). The presence of large clades of closely related OTUs (Fig 3) limits our ability to compare the results of the RT and the BM models but results do show greater similarity of growth traits among closely related fast-growing taxa (e.g., within Hypocreales), than among closely related slow-growing taxa (e.g., within Xylariales).

## Discussion

Our results demonstrate previously undescribed and cryptic functional diversity in resource use and growth of important plant-associated symbionts, the fungal endophytes. We evaluated the taxonomic composition, phylogenetic diversity, and resource use and growth traits of endophyte communities associated with *A. gerardii* leaves to obtain three prominent results. First, we discovered that *A. gerardii* endophyte communities are composed of slow- and fast-growing assemblages that were differentially recovered by leaf sectioning and leaf maceration isolation methods. Second, fungi in these two assemblages differ in their niche width as the slow-growing fungi exhibited narrower but more varied niche widths than did the fast-growing fungi. Third, while the slow- and fast-growing assemblages are phylogenetically distinguishable at the level of fungal class and order, patterns of resource use and growth varied at each phylogenetic level from species to phylum.

The mean standardized growth of isolates sorted into distinct slow- and fast-growing assemblages, and these were differently recovered by the two isolation methods (Fig 1, S3 Fig). The two isolation methods apparently favor different fungal growth types; the sectioning approach obtained more slow-growing fungi and the leaf maceration approach obtained more fast-growing fungi. While the cause for the differing recovery of growth types is not easily discerned, the recovery of two assemblages exhibiting distinct growth rates was unexpected as they have not previously been reported in similar studies (e.g., [3, 12, 43, 68, 89]). More

importantly, the slow-growing and fast-growing assemblages are associated with differing patterns of resource use as taxa in the slow-growing assemblage exhibited a lower mean niche width, but greater variation in resource use than fungi in the fast-growing assemblage (Figs 2 and 3). Differences in growth patterns among co-occurring taxa have been reported for other fungal guilds as well [90–92]. For example, the foraging behaviors of saprobic fungi can be classified into faster-growing "guerrilla" phenotypes with directional growth toward a resource, and slower-growing "phalanx" phenotypes with a broad colony growth front less directed toward a resource [90, 93, 94]. In mycorrhizal fungi, long-distance exploration phenotypes tend to sequester greater amounts of nitrogen and carbon than do taxa exhibiting short-distance exploration phenotypes [95, 96]. Together, the results of our and other studies of symbiotic fungal communities demonstrate functional diversity in resource use and growth that will remain cryptic until assessed with living cultures [97].

Ancient and ongoing evolutionary processes generated the diversity of resource use and growth traits that we observe. The deep evolutionary origins of taxa constituting slow- and fast-growing assemblages is demonstrated by their distinction at the class and order level (Fig 3), divergences encompassing over 120–300 million years [98–101]. Within the Sordariomycetes, a fungal class often represented in endophytic communities of grasses [12, 65, 68, 89, 102–104], differences in growth traits characterize taxa belonging to different orders (Fig 3). While our ability to generalize to endophyte communities of all plants is limited by our small sample of *A. gerardii* leaves, deep evolutionary origins of functional traits have previously been demonstrated in endophytic fungi [41, 42, 105] and other fungal symbionts of plants as well [106–110]. For example, Maherali and Klironomos [111] found that divergence in hyphal growth traits of mycorrhizal fungi were often at the level of order and family. The evolution of multiple lineages of slow- and fast-growing taxa within the phylum Ascomycota has left a legacy of diversity in resource use and growth traits exhibited in each sampled leaf and across the experimental site. Nonetheless, our results also point to the contributions of ongoing evolutionary processes to the observed diversity. We demonstrate slow- and fast-growing fungi among closely related taxa across the phylogeny (Fig 3) and within the Dothideomycetes, which are commonly isolated in endophytes studies [10, 44]. The variation in resource use and growth traits demonstrated within endophyte communities of *A. gerardii* and other plant hosts [63, 105, 112] suggests a rich source of functional diversity that may be deployed to improve resiliency of this iconic prairie grass against the effects of climate change.

Because *A. gerardii* leaves emerge, expand, and senesce within a growing season, their endophytic communities are transient and likely structured by both stochastic and deterministic processes occurring at differing spatial scales. We observe appreciable similarity in resource use among fungi that co-occur within leaves, consistent with high levels of functional redundancy observed in many microbial symbiont communities [113, 114]. That functional redundancy may be maintained by stochastic processes as endophytic communities are largely established by dispersal and colonization from sources at larger scales than the individual plants or plots [12, 68, 115, 116] and from heterogeneous environmental sources [73, 117–120]. These fungi largely reproduce as saprobes, rather than within living plant hosts [58, 121], and the many differences among taxa in the two assemblages may have evolved as a consequence of interactions in the environments in which these organisms reproduce rather than within leaves [120]. Although competition within the host [45, 122–126] likely also plays a role in the evolution and maintenance of variation in functional traits that we observe, our ability to infer processes occurring *in vivo* from *in vitro* results are limited. Nonetheless, we observe substantial functional variation among closely related fungi, the causes for which await population level experimental analysis of ecological and evolutionary processes across soil and host environments. The co-occurrence of fungi belonging to slow- and fast-growing assemblages

within each sampled leaf no doubt reflects the consequences of eco-evo processes occurring over varied spatial and temporal scales to generate and maintain the cryptic but pervasive coexistence of two functionally different assemblages within the fungal endophyte communities of *A. gerardii*.

## Conclusion

Our study reveals that distinct slow- and fast-growing endophyte assemblages inhabit leaves of the grass *A. gerardii*. While many differences in resource use and growth traits that characterize these assemblages have their origins deep in the Ascomycete phylogenetic history, resource use and growth traits are also continuously evolving. Together these results suggest that endophyte functional diversity is likely maintained by both stochastic processes such as dispersal from heterogeneous environments, and more deterministic processes such as competition and niche partitioning in the environments in which the fungi reproduce. In addition, the high levels of variation in functional traits among closely related taxa that we observe here suggest caution in assigning functional traits from taxon assignments based on barcode sequences alone. Our results show that using different culture isolation approaches, as well as deploying ecological and evolutionary analytical and experimental approaches will substantially improve our understanding of the functional and taxonomic diversity of symbiotic microbiomes.

## Supporting information

**S1 Fig. Phylogeny of all fungi isolated by leaf sectioning and maceration.** Species names assigned to isolate are indicated at the tips of tree and are color-coded by isolation method (sky-blue: leaf sectioning, dark purple: leaf maceration). Adjacent columns represent fungal class and order ("Other" represents fungal classes or orders with a relative abundance $\leq$ 2%). (JPG)

**S2 Fig. Frequency distributions of isolates using differing numbers of Biolog carbon resource.** Graph shows the percent of fungal isolates that use (standardized growth > 0.005 $OD_{590}$) a given number of Biolog carbon resources. Of the 240 isolates evaluated for use and growth on Biolog carbon resources, 50% of isolates (red dotted line) obtained by leaf sectioning (**A**) used 62 Biolog carbon resources (red arrow). For isolates obtained by leaf maceration (**B**), 50% used 85 Biolog carbon resources. (TIF)

**S3 Fig. Distribution of mean standardized growth values in slow- and fast-growing isolates.** Mean standardized growth of fungal isolates calculated using three different metrics demonstrated bimodal distributions. Mean standardized growth was calculated on the basis of 62 Biolog carbon resources (**A**), all 95 resources (**B**) and only those resources used by each individual isolate (**C**). Cut-off values derived from finite mixture model fitting ([87], A; 0.12, B; 0.12, C; 0.1) distinguishing slow- (blue bars) and fast-growing (gold bars) assemblage are shown (black arrow). (TIF)

**S1 Table. Use of 95 Biolog resources (standardized growth > 0.005 $OD_{590}$) by fungal isolates recovered from leaf sectioning and maceration isolation methods.** For each of the 95 Biolog carbon resources, the percent of isolates obtained by each isolation method and using that resource are indicated. Resources are rank in order from most often used (1) to least often used resources by both sets of isolates. Mean standardized growth of each isolate was calculated based on standardized growth on the 62 resources used by at least 50% of the isolates

(ranks in bold).
(DOCX)

**S2 Table. Taxonomic assignment of OTUs.** For each OTU, the assemblage (fast, slow, or shared in both assemblages), Phylum, Class, Order, and Genus/species of best taxon matches are reported. GenBank Accession numbers of reference sequences used to assign taxonomy using the Evolutionary Placement Approach (EPA) within TBAS [77, 78] are given in the last column.
(DOCX)

**S3 Table. Phylogenetic structure of fungi belonging to slow- and fast-growing assemblages.** For each fungal class or order, the number of isolates that belong to either the fast- (mean standardized growth $> 0.12$ $OD_{590}$) or slow-growing (mean standardized growth $< 0.12$ $OD_{590}$) assemblage, and totals, are given. Equality of the number of slow- and fast-growing isolates assigned to each taxonomic group was evaluated using a pairwise Fisher exact test (NS, not significant at $P < 0.05$; ***: $P < 0.001$). Pairwise comparisons could not be made for groups with $n \leq 5$ (NA). P-values were adjusted for multiple comparison using the Holm–Bonferroni method (Adjusted p-value; rowwise_fisher_test function of the rstatix package [88]).
(DOCX)

**S4 Table. Results of phylogenetic analysis for growth on individual Biolog carbon resources.** Conservation of standardized growth on each of the 95 Biolog carbon resource was assessed using Blomberg's K statistic [61] under the Brownian motion (BM) and Random Tip (RT) models and estimated for the entire phylogeny, and for the most common fungal classes, and most common orders within the class Sordariomycetes in the sample. Under the BM model, K statistics greater than 1 ($K > 1$; bold) indicates that traits are more conserved than expected by chance. Under the RT model, *$P < 0.05$ and **$P < 0.01$ indicates K significantly greater than expected by chance. Total number of resources for which standardized growth was more conserved than expected under null BM or RT models are reported on the last rows and these values are reported in Table 1.
(DOCX)

**S1 Data. Metadata of all fungi isolated by leaf sectioning and maceration.**
(CSV)

**S2 Data. Raw resource use data of fungi evaluated on Biolog plates.**
(CSV)

## Acknowledgments

We thank past and current members of the May and Kinkel Lab at the University of Minnesota for sample collection, culture isolation, and DNA sequencing, and many discussions of the conceptual basis of the study. We also thank the staff and interns at Cedar Creek LTER and the Nutrient Network (NutNet) for providing resources that contributed to results reported in this paper. We are grateful to Jeannine Cavender-Bares and Jesús N. Pinto-Ledezma for fruitful discussions on the conceptual framework of community phylogenetic analyses. We thank Mara Demers, Monica Watson, Emma Daily, George Furey, Michael Travisano, Ruth Shaw, and Marlene Zuk for their valuable comments and suggestions on earlier drafts of the manuscript. The use of any trade, firm, or corporation names in this publication does not constitute an official endorsement or approval by the U.S. Department of Agriculture or the Agricultural Research Service. The findings and conclusions in this publication are those of the authors and

should not be construed to represent any official USDA or U.S. Government determination or policy. USDA is an equal opportunity provider and employer.

## Author Contributions

**Conceptualization:** Cedric Ndinga-Muniania, Elizabeth T. Borer, Eric W. Seabloom, Linda Kinkel, Georgiana May.

**Formal analysis:** Cedric Ndinga-Muniania, Nicholas Wornson.

**Funding acquisition:** Elizabeth T. Borer, Eric W. Seabloom, Linda Kinkel, Georgiana May.

**Resources:** Linda Kinkel, Georgiana May.

**Software:** Cedric Ndinga-Muniania, Nicholas Wornson, Michael R. Fulcher.

**Supervision:** Georgiana May.

**Validation:** Cedric Ndinga-Muniania.

**Visualization:** Cedric Ndinga-Muniania.

**Writing – original draft:** Cedric Ndinga-Muniania.

**Writing – review & editing:** Michael R. Fulcher, Elizabeth T. Borer, Eric W. Seabloom, Linda Kinkel, Georgiana May.

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
