## [Decision Letter · Decision Letter 0]

5 May 2023

PONE-D-23-10504Cryptic functional diversity within a grass mycobiomePLOS ONE

Dear Dr. Ndinga-Muniania,

Thank you for submitting your manuscript to PLOS ONE. After careful consideration, we feel that it has merit but does not fully meet PLOS ONE’s publication criteria as it currently stands. Therefore, we invite you to submit a revised version of the manuscript that addresses the points raised during the review process.

We look forward to receiving your revised manuscript.

Kind regards,

Tzen-Yuh Chiang

Academic Editor

PLOS ONE

Journal Requirements:

Reviewers' comments:

Reviewer's Responses to Questions

**Comments to the Author**

1. Is the manuscript technically sound, and do the data support the conclusions?

Reviewer #1: Yes

2. Has the statistical analysis been performed appropriately and rigorously? 

Reviewer #1: Yes

3. Have the authors made all data underlying the findings in their manuscript fully available?

Reviewer #1: Yes

4. Is the manuscript presented in an intelligible fashion and written in standard English?

Reviewer #1: Yes

5. Review Comments to the Author

Reviewer #1: Andropogon gerardii leaf associated fungal communities were examined using two different methods of isolating - either sectioning or macerating the leaves. Surprisingly or not, the two different isolation methods yielded taxa representing different lineages. The acquired isolates were evaluated for their C use on the BIOLOG system. A collection of isolates was divided into fast- and slow-growing fungi by a cut off OD in the BIOLOG plates and the similarity of observed functional traits were used to estimate their distribution over evolutionary history. Further, the fast- and slow-growing isolates’ niche width is estimated by number of C-substrates among the 95 that were available on the BIOLOG plates. These analyses highlight the difference between the observed slow- and fast-growing isolates with lower mean but higher variance among the former isolates.

The contribution arrives to three conclusions: 1) that A. gerardii endophytes include fast and slow growing members; 2) that the slow and fast-growing isolates differ in their niche breadth and variability; 3) that these differences may be conserved or deep-rooted in phylogenies as the assemblages could be distinguished on high levels of taxonomic hierarchy. While #1 may be rather expected, #3 is intriguing and may stimulate some further research to confirm. Whether supported or not is irrelevant as the conclusion is quite thought-provoking and supported by the limited data available from the reported studies. The authors conclude that the communities include undescribed, functionally and phylogenetically distinct fungi providing insight into the stochastic/deterministic processes underlying the assembly of foliar fungal communities.

In more general terms, the reported research all falls in line with widespread functional diversity within fungal guilds and that - although the traits (C-use) are variable - they are conserved within lineages.

I have a few general concerns and some minor suggestions. First, although perhaps based on a small sample size (12 leaves and 240 random isolates for the analyses), the analyses are meticulous and lead to well-justified conclusions. However, I ask authors to acknowledge the caveat of small sample size in generalizing their conclusions. Second, I am uncertain if metabarcoding is appropriate to use here. Perhaps this is more in line with ITS barcode identification of acquired isolates. Thirds, does the melanization of the fungal hyphae affect the OD, therefore potentially resulting in grouping of light and dark cultures erroneously?

MINOR COMMENTS:

Line 60: “important but diverse” - Maybe important and diverse would be better?

Line 67: need

Line 102: often dominate. Remove “often”

Line 122: Additional background information on the age and management of the grass is necessary. The leaf samples were collected in August after flowering time - would this effect fungal endophytes?

Line 122: Some treatment plots were treated with the nitrogen fertilizer. Although perhaps beyond the scope of the current study, where there any effect of this treatment of the fungal communities?

Lines 137-138: It is unclear how the fungal colonies were collected from 1.5mL Eppendorfs. It seems that fast growing endophytes would take over the limited space and it would be hard to isolate slow-growing isolates?

Lines 143–144: More details are needed on the isolation process. How were single taxon pure cultures confirmed?

Line 145: Can we make conclusions about the growth of the fungi on artificial media such as 2% PDA for 6 months? As stated earlier in the text endophytes are plant symbionts, how can we ensure the same functional behavior of the fungi on the media without the plant symbiont?

Line 173: This assumes that all cultures sporulated. Were the isolates screened for conidia before selecting them for further assays.

6. PLOS authors have the option to publish the peer review history of their article (what does this mean?). If published, this will include your full peer review and any attached files.

Reviewer #1: No

---

## [Author Response · Author response to Decision Letter 0]

12 Jun 2023

Responses to the Academic Editor

Respose: We made changes to our funding information and wish our Financial Disclosure to read as follows:

This study was supported by a National Science Foundation (NSF) Macrosystems Biology grant (NSF-DEB 00037623) to co-PIs EB, ES, LK, GM). Support was also provided from the NSF Long Term Ecological Research (NSF-DEB-1234162 and NSF-DEB-1831944 to Cedar Creek LTER) and Research Coordination Network (NSF-DEB-1042132) programs. Support to CNM was provided by the NSF-DEB, a Dissertation Fellowship from the Graduate School at University of Minnesota, and from the Graduate Program in Plant and Microbial Biology. The funders had no role in study design, data collection and analysis, decision to publish, or preparation of the manuscript.

Responses to Reviewers' comments

Review Comments to the Author:

Reviewer #1: Andropogon gerardii leaf associated fungal communities were examined using two different methods of isolating - either sectioning or macerating the leaves. Surprisingly or not, the two different isolation methods yielded taxa representing different lineages. The acquired isolates were evaluated for their C use on the BIOLOG system. A collection of isolates was divided into fast- and slow-growing fungi by a cut off OD in the BIOLOG plates and the similarity of observed functional traits were used to estimate their distribution over evolutionary history. Further, the fast- and slow-growing isolates’ niche width is estimated by number of C-substrates among the 95 that were available on the BIOLOG plates. These analyses highlight the difference between the observed slow- and fast-growing isolates with lower mean but higher variance among the former isolates.

The contribution arrives to three conclusions: 1) that A. gerardii endophytes include fast and slow growing members; 2) that the slow and fast-growing isolates differ in their niche breadth and variability; 3) that these differences may be conserved or deep-rooted in phylogenies as the assemblages could be distinguished on high levels of taxonomic hierarchy. While #1 may be rather expected, #3 is intriguing and may stimulate some further research to confirm. Whether supported or not is irrelevant as the conclusion is quite thought-provoking and supported by the limited data available from the reported studies. The authors conclude that the communities include undescribed, functionally and phylogenetically distinct fungi providing insight into the stochastic/deterministic processes underlying the assembly of foliar fungal communities.

In more general terms, the reported research all falls in line with widespread functional diversity within fungal guilds and that - although the traits (C-use) are variable - they are conserved within lineages.

Response: Thank you! We also hope that our results will stimulate further research. As we acknowledge below, while the sampling is based on relatively few leaves (12), the observations are consistent across these leaves, and reporting this new observation of functionally distinct endophyte assemblages will spur further research. 

Reviewer #1: I have a few general concerns and some minor suggestions. First, although perhaps based on a small sample size (12 leaves and 240 random isolates for the analyses), the analyses are meticulous and lead to well-justified conclusions. 

Response: Thank you, we did take great care in the analyses and distinguishing conclusions justified from the results, and inferences such as those regarding assembly processes. 

Reviewer #1: However, I ask authors to acknowledge the caveat of small sample size in generalizing their conclusions.

Response: We agree. While we did sample a fairly large number of isolates for functional analyses (240), at lines 417-419, we now acknowledge that sampling only 12 leaves (3 plants from each of 4 plots) limits our abilities to generalize to endophytic communities generally.

Reviewer #1: Second, I am uncertain if metabarcoding is appropriate to use here. Perhaps this is more in line with ITS barcode identification of acquired isolates. 

Response: We understand that “metabarcoding” is a term used in many ways and thus potentially confusing. However, there may be a misunderstanding here because we do not use metabarcoding approaches (community sequencing) or terminology. We used Sanger sequencing of the fungal ITS barcode. We made edits in lines 152-160 that we hope will clarify our methods.

Reviewer #1: Thirds, does the melanization of the fungal hyphae affect the OD, therefore potentially resulting in grouping of light and dark cultures erroneously? 

Response: We had not considered this and appreciate the question. The more melanized fungi such as those in the order Xylariales were most often also slow-growing, which is contrary to what we would expect if melanization upwardly biased the OD readings. We have now included that information at lines 303-305.

MINOR COMMENTS:

Line 60: “important but diverse” - Maybe important and diverse would be better?

Response: Done.

Line 67: need

Response: Done.

Line 102: often dominate. Remove “often”

Response: As A. gerardii is not dominant across the whole region of the Great Plains, we edited the text at lines 101-102 for more clarity.

Line 122: Additional background information on the age and management of the grass is necessary. 

Response: We provided edits for clarity on the management of the experimental NutNet site (lines 118-129). In short, it was established in 2007 and nutrient treatments have been applied annually since then.

---

## [Editor Report · Decision Letter 1]

19 Jun 2023

Cryptic functional diversity within a grass mycobiome

PONE-D-23-10504R1

Dear Dr. Ndinga-Muniania,

We’re pleased to inform you that your manuscript has been judged scientifically suitable for publication and will be formally accepted for publication once it meets all outstanding technical requirements.

Kind regards,

Tzen-Yuh Chiang

Academic Editor

PLOS ONE
---

## [Editor Report · Acceptance letter]

12 Jul 2023

PONE-D-23-10504R1 

Cryptic functional diversity within a grass mycobiome 

Dear Dr. Ndinga-Muniania:

I'm pleased to inform you that your manuscript has been deemed suitable for publication in PLOS ONE. Congratulations! Your manuscript is now with our production department. 

Kind regards, 

on behalf of

Dr. Tzen-Yuh Chiang 

Academic Editor

PLOS ONE